

# Architecture of Solution for Panoramic Image Blurring in GIS projects Application

Dejan Vasić[1], Marina Davidović [1*], Ivan Radosavljević[2] and Đorđe Obradović[2]

[1]Department for civil engineering and geodesy, Faculty of Technical Sciences, University of Novi Sad, Novi Sad, 21000, Vojvodina, Republic of Serbia
[2]School of Informatics and Computing, Singidunum University, Belgrade, 160622, Republic of Serbia;

*Correspondence to*: Marina Davidović (marina.davidovic@uns.ac.rs)

**Abstract.** Panoramic images captured using laser scanning technologies, which principally produce point clouds, are readily
applicable in colorization of point cloud, detailed visual inspection, road defect detection, spatial entities extraction, diverse maps creation etc. This paper underlines the importance of images in modern surveying technologies and different GIS projects at the same time having regard to their anonymization in accordance with GDPR. Namely, it is a legislative requirement that faces of persons and license plates of vehicles in the collected data are blurred. The objective of this paper is to present a novel architecture of the solution for a particular object blurring. The methodology was tested on four data sets
counting 5000, 10 000, 15 000 and 20 000 panoramic images respectively. Percentage of accuracy, i.e. successfully detected and blurred objects of interest, was higher than 97% for each data set.

**Keywords.** blurring, feature detection, image processing, mobile mapping, geoinformation, panoramic images

## 1 Introduction

The rapid advancement of the technologies used in geodesy and geomatics has opened up many possibilities in various
scientific spheres. A tremendous development of information technology is one of the driving factors behind that great growth of surveying and geodesy science (Habib et al., 2020). The implementation of laser scanning technology combined with a high precision navigation system enables 3D scanning of road infrastructure (Mobile laser scanning – MLS). By using this method, time consumption is reduced (Sztubecki et al., 2020) and it is possible to obtain a significantly larger amount of information. Furthermore, the representation of a larger terrain area with a higher level of detail is enabled, as well as various
additional analyses and high-efficiency processing of the collected data.

Reliable feature extraction from 3D point cloud data is an important phase in numerous application domains, such as traffic managing, object recognition, autonomous navigation, civil engineering and architectural projects, etc. All those projects require reliable, quality data clearly demonstrating the real-time conditions in the field to be successfully completed. When surveying the terrain with a MLS, besides a point cloud as the main output product, the digital camera collects panoramic
photographs. Each photo is associated with the appropriate position from the trajectory and in this way the photo is matched with the point cloud (Batilović et al., 2019). Those images are gaining in importance due to the processes of visualization

and different maps creation. With image usage, visual inspection is facilitated and various types of damage can be detected. According to (Lahoti et al., 2019), maps that graphically present particular data of interest about an urban area can be generated. That has been shown very helpful for planners and architects in positioning future objects, regulating green areas,
traffic management, forest management (Kuzmić et al., 2017), and many other. Since images play a vital role in a lot of different disciplines, they have inevitably become part of collected data. These most often involve cars and pedestrians. Following GDPR (European Data Protection Regulation), those pictures are to have blurred faces of people and car license plates. In order to secure legitimate panoramic images, the detection and blurring of the above-mentioned features ought to be conducted. Whilst it is clear that there are justifiable reasons for sharing multimedia data acquired in such ways (e.g. for
law enforcement, forensics, bioterrorism surveillance, disaster prediction), there is also a strong need to protect the privacy of innocent individuals who are inexorably "captured" in the recordings (Ribaric et al., 2016). For instance, the average citizen in London is caught on CCTV cameras about 300 times a day (Cavalaro 2007).

Moreover, the necessity for object detection has significantly increased. The reasons for this include a growing demand for automatic vehicle identification required for traffic control, border control, access control, calculation of parking time and
payment, search for stolen cars or unpaid fees, along with the requirement for reliable identification considering a complex diversity of circumstances, e.g. different lighting conditions, presence of random or structured noise in the plate, its size and type of characters as well as nationality specific features (Kasaei et al., 2010).

This study focuses on automatic object detection from panoramic images, obtained by mobile mapping technology, which is followed by blurring of those objects.
The remainder of the paper is divided into four sections. Section Related works provides a descriptive summary of certain methods that have been implemented and tested in the area of automatic object detection and blurring. Section Materials and Methods offers an insight into the proposed methodology. Experimental results are discussed in Section Results and Discussion. Conclusions and further work are presented in the last section.

## 2. Related works

Some studies related to the paper topic are presented in this section. Several authors define object detection and its significance (Demir 2014, Božić-Štulić et al., 2018, Radović et al., 2017). Some authors deal with object detection in general, while the others point out the detection of particular spatial entities. In recent years, deep learning approaches using features extracted by convolutional neural networks (CNN) have significantly improved the detection accuracy. Paper (Sommer et al., 2017) proposes a deep neural network derived from the Faster R-CNN approach for multi-category object
detection in aerial images. The detection accuracy was shown to be capable of improvement by replacing the network architecture with the one specially designed for handling small objects. Faster R-CNN approach for medium-sized objects was elaborated in paper (Zhang et al., 2016). Authors (Božić-Štulić et al., 2018) used a pre-trained Faster R-CNN model for detection of minor deformations from images obtained by UAV surveying technology. The article (Radović et al., 2017)



details the procedure and parameters used for the training of CNNs on a set of aerial images for object recognition. The results show that by selecting a proper set of parameters, CNN can detect and classify objects with a high level of accuracy (97.5%) and computational efficiency.

According to (Deb and Jo, 2009), the vehicle license plate detection from vehicle images is a challenging task due to multi-style plate formats, viewpoint changes and the non-uniform outdoor illumination conditions during image acquisition. A real-time multiple license plate detection algorithm is described in (Asif et al., 2016). The authors used color components to identify license plate regions. Experimental results show that the proposed method accurately detects 93.86% of these elements. Edge feature-based method uses edge detection and morphological operations to find a rectangular candidate plate and then aspect ratio to filter the candidate regions. While this approach can work in many cases, authors of (Chuang et al., 2014) showed that skewed plates and small plates cannot be detected. Another research (Hamid and Shayegh, 2013) advises the use of edge detection and morphological operations to identify potential license regions followed by connected components operation to identify the license plate location. Although the correct recognition rate was reported to be 98.66%, this method requires multiple steps. Namely, the obtained image needs to be converted into a binary mode first and only then the algorithm is conducted. Also, the acquired time from input until final output is not mentioned. Authors (Wang and Lee, 2003) detected probable license plate regions from the gradients of the input car images. Then, this element was separated into several adjacent regions and the one with the largest possible value was chosen. Experimental results show that the rotation-free character recognition method can achieve an accuracy rate of 98.6%. The flow of the suggested algorithm was the manual detection of character features that are non-sensitive to rotation variations. Region-based license plate detection method was described in (Jia et al., 2007), where a mean shift procedure is applied in a spatial-range domain to segment a color vehicle image in order to get candidate regions. License plates adhere to a unique feature combination of rectangularity, aspect ratio and edge density. These three features are defined and extracted in order to decide if a candidate region contains this object. The lack of this method is the difficulty of detecting license plates in case vehicles and their respective plates are of similar or same color.

Human body detection presents a number of challenges such as extracting meaningful features to capture a wide range of poses of human appearance (Deb and Jo, 2009). Most current work on human detection in color images encompasses a variety of feature descriptors and classifiers. Most notable people detection method is based on the histogram of oriented gradients (HOG) feature descriptors (Dalal and Triggs, 2005). Dense descriptors comprising blocks with multiple histograms of image gradients are classified as human/non-human using a linear SVM. The histogram of image gradients is constructed and scaled for each cell. The final search window descriptor is a vector of concatenated block histograms. Although this method has been proven as reasonably efficient, there is still room for optimization and further speed-up in detections. The paper (Miezianko and Pokrajac, 2008) documented a method for detecting people in low-resolution infrared videos. The suggested method is based on extracting gradient histograms from recursively generated patches and, subsequently, computing histogram ratios between the patches. Each set of patches was defined in terms of relative position within the search window, and each set was then recursively applied to extract smaller patches. The major objective of this study was to

incorporate motion detection and tracking into the existing system and to limit the search in the future. Another publication (Breckon et al., 2012) gave an account of a combined autonomous system for surveillance and human detection, which can

also be applied to vehicle detection, using optical and thermal images. This approach primarily detects the initial segments within the scene that might contain an object. Afterwards, isolated segments are extracted, supplying a basis for secondary object classification to be carried out. As this method does not take place in real-time, authors of publication (Gilmore et al., 2011) suggested almost real-time detection algorithm, founded on digital, infrared thermal imagery. The objective was to achieve pedestrian candidate selection and detection. The focus of the article (Vu et al., 2006) is an event recognition system

employing face detection and tracking combined with audio analysis. Three-dimensional contexts, such as zones of interest and static objects, were recorded in a knowledge base and 3D positions were calculated for mobile objects using calibration matrices. The major flaw of this mechanism is the fact that substantial changes in lighting conditions occasionally prevent the system from detecting people correctly.

Nowadays, there is a growing trend in image blurring so various authors have paid considerable attention to research this

area. Authors of (Farid et al., 2018) described content-adaptive blurring (CAB). In the CAB, a multi-focus image is iteratively blurred in such a manner that only the focused regions get blurred whereas the defocus regions receive a little or no blur at all. If blurring degrades the quality of a local image region more than an allowable limit, that region is not blurred and exempted from further blurring. Thus, the defocused regions are preserved while the focused regions get blurred. In (Chiang et al., 2016) different image types were exploited for multiple object recognition by means of focusing and blurring.

The focusing and blurring step applies image processing techniques to focus on the most important objects and blur out the rest of the image with either vignette, blur, or bokeh, using the identified object bounding boxes. A method named inhomogeneous principal component blur (IPCB) was proposed in (Du et al., 2011). It adaptively blurs different pixels of a license plate by taking into account the prior distribution of sensitive information. The blurring is based on the Principal Component Analysis (PCA) approach – the original plate's area is substituted by a reconstructed area that is obtained by

applying a smaller number of eigenvectors. The detection of faces and license plates in Google Street View footage was demonstrated in (From et al., 2009), where de-identifications were simply done by blurring the detected locations. A simplified version of the face detector based on a fast sliding-window approach over a range of window sizes was used for the detection of license plates. They belong to a large family of sliding window detectors, such as Schneiderman and Kanade (2001) and Viola-Jones detectors (2004). The authors have reported that a completely automatic system detected and

sufficiently blurred 94-96% of the total number of license plates and more than 86% of faces in evaluation sets sampled from Google Street View imagery.

## 3 Materials and Methods

The solution proposed in this paper is composed of four software components:

- Project management module;





- Vehicles and people detection module;
- License plate detection module;
- Blurring module.

The project management module is a web application that provides a user interface (UI) necessary to coordinate the blurring process. Through this interface, it is possible to setup and monitor the blurring process and manually correct the obtained

results. Besides the UI, the project management module is responsible for distribution of configurations and tasks to the rest of the modules. It is also the main collection point for the results produced by the aforementioned modules and the final processing unit which encompasses the blurring module. The backend of the project management module was implemented using the Flaks web application framework (The Pallets Projects), ZeroMQ and the OpenCV library (ZeroMQ, Open Source Computer Vision Library). The backend exposes two ZeroMQ TCP client sockets which are utilized to send processing

requests and to receive responses from other modules. The first socket is used to communicate with the vehicles and people detection module, whilst the second one is intended for the communication with the license plate detection module. The front-end of the project management module was implemented using the Angular framework and OpenLayers library (Angular, OpenLayers).

The vehicle and people detection module and the license plate detection module are responsible for the extraction of the

bounding boxes from the raw images supplied by the project management module. These modules can contain one or more processing nodes. Each processing node exposes a TCP server ZeroMQ socket. The server sockets accept requests from the project management module. Each request contains the actual image that has to be processed. After its processing, the same socket is used to reply to the project management module with a message containing the bounding boxes extracted from the input image.

The processing nodes of the vehicle and people detection module have been implemented as a neural network trained for object detection. This network was implemented based on the TensorFlow Object Detection API (Git Hub). The model used from the API was the Faster R-CNN ResNet-101 trained on the COCO data sets. This model was trained to detect all of the 80 classes in the COCO data set which include vehicles and people. The output of this neural network consists of the bounding boxes and classes of the detected objects. Although the input into the processing node is the whole image, the input

into the network comprises overlapping patches of the original image scaled down to the resolution of 1000 by 600 pixels. The minimum overlap of the patches is 50% which ensures that no objects are skipped or partially detected. The main advantage of splitting the image into smaller patches is the reduction of the required computational resources. The downside of this approach is the creation of multiple overlapping bounding boxes for the same detected object. Once returned to the project management module, these overlapping boxes have to be merged into a single bounding box. The process of merging

was done using the non-maximum suppression algorithm (Neubeck and Van Gool, 2006). The processing nodes of the license plate detection module were implemented by the means of the same model as those of the vehicle and people one. However, the neural network used in the license plate detection nodes was additionally trained to detect license plates in the images of vehicles. The training was conducted on a data set containing 1,000 examples of vehicle images which contained a





license plate. Each license plate was supplied with both the image and its bounding box. The bounding boxes and paths of

the images they belonged to were stored in Pascal VOC xml format. The number of training examples was chosen empirically and proven sufficient for the neural network to learn distinctive patterns of license plates. The input into the license plate detection module consists of the images of vehicles provided by the project management module. These were generated by the project management module from raw images by extracting patches corresponding to the bounding boxes of vehicles detected in the vehicle and people detection module. The output of this module comprises the bounding boxes of

license plates detected on the input patches with their coordinates adjusted with respect to the original image.

The blurring module is an integral part of the project management module and is responsible for blurring the areas designated by both the vehicle and people detection module and the license plate detection module. The input into this module represents raw images and the bounding boxes obtained by processing images in the aforementioned modules. A set of bounding boxes, detected in the same image, was used for each corresponding input image to determine the area of the

image that has to be blurred. In case the bounding box corresponds to a license plate, the area of the image covered by the bounding box is immediately blurred using the Gaussian blur. However, if the bounding box corresponds to a person, the blurring module first detects the face of that person and then blurs the area of the detected face using the Gaussian blur. The idea was to avoid unnecessary blurring of faces on billboards and similar public displays — the ones that are recognized as faces but they do not belong to the group prone to the risk of identity misuse or are not considered sensitive personal

information. This kind of procedure aims to facilitate the processing performed by the algorithm since it does not require search of all images, but just the detected areas. This makes the methodology described in this paper faster and more precise. The face detection was done using the Haar classifier provided by the OpenCV library. The classifier was trained to detect faces from the frontal view. The output of the blurring module presents blurred images and bounding boxes of the blurred areas. These two elements are stored in the output folder defined in the project management module.

An overview of the proposed innovative architecture of the solution as well as the data flow between components is shown in Figure 1.

**Figure 1: Architecture of the solution**

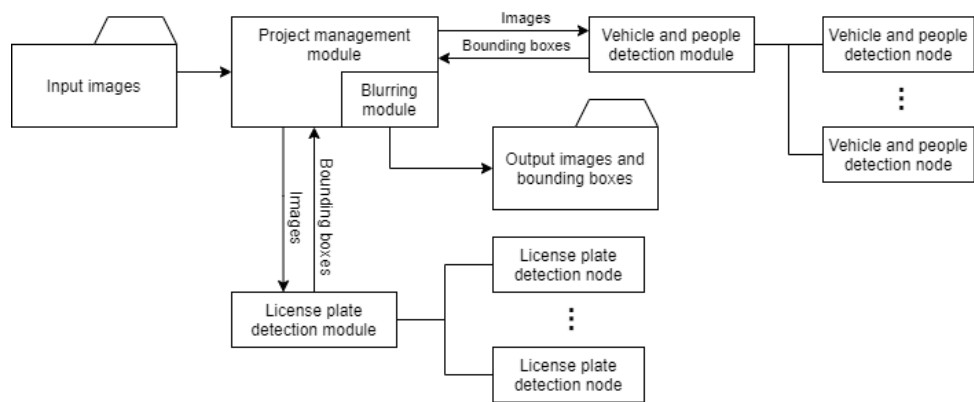





The main advantage of the architecture of the solution elaborated in this paper is the ease with which it can be scaled up in
order to reduce the computation time necessary to process large amounts of images. This scaling can be done by deploying
additional processing nodes. Each node can be deployed on a dedicated computer or it can share a single computer with
other nodes. If the nodes are used on dedicated computers, it is necessary to connect those computers in a computer network
to allow them to share data. Due to the amount of data that is shared between the nodes during the processing, it is necessary
to provide a network infrastructure with bandwidth of no less than 1Gbit/s to achieve an optimal performance of the solution.
Also, the nodes were implemented to utilize CUDA capable GPUs to increase the speed of both training and exploitation of
the model. Thus, each node requires a dedicated CUDA capable GPU in order to be successfully used.

## 4. Results and Discussion

As mentioned in the Introduction, the need for images in mobile mapping projects is being increasingly recognized over
time. Nowadays, these projects have put forward high demands regarding precision together with the level of details and
accuracy. Therefore, producing non-colorized point cloud frequently cannot satisfy all these requirements. Colorization of
the point cloud is performed with the help of collected images and this has helped a more efficient extraction and element
recognition. Additionally, all road defects like cracking, potholes, patching, surface chip loss, and other can easily be
managed with image usage. Often, the results of mobile mapping projects are published on public servers. Matching the
photographs to point cloud and publishing them on a web-platform offers a significant advantage. This convenience is
reflected in the fact that while looking at the point cloud, photographs can be observed at the same time allowing any
possible doubts about the particular terrain situation to be solved (Batilović et al., 2019). In order to use images for the
above-discussed purposes, they have to be GDPR-compliant, i.e. blurring of faces and license plates is required to ensure
privacy protection. Consequently, the suggested mechanism for object detection and blurring has a substantial impact on
diverse projects.
This section focuses on testing the methodology explained in the previous chapter. In order to verify the validity and
accuracy of the suggested solution architecture, several experiments were carried out.  Four data sets containing 5000, 10
000, 15 000 and 20 000 images respectively were used in each trial. They represent a collection of panoramic images with
the resolution of 8000 x 4000 pixels, obtained by mobile mapping scanning with Trimble MX9 system. The fact that
people's faces and license plates are taken at a different angle and have divergent size and position in panoramic images has
made this experiment more challenging. Additionally, the project does not encompass uniform – but different — vehicle
types: cars, trucks, vans, motorcycles etc. The computer performances used for conducting these tests are presented in Table
1, while their results are given in Table 2.

**Table 1. Specifications of computer used for experiment**

| Component | Type |
| --- | --- |





| CPU | 2nd Gen Ryzen™ 7 2700X Desktop Processor -- 16T @ 3.7GHz |
|---|---|
| RAM Memory | 2 x Kingston HyperX Predator CL16, HX432C16PB3/16 -- 2 x 16GB @ 3200MHz |
| Graphic card | 2 x Nvidia GeForce GTX 1660 SUPER -- 6GB |
| SSD | Crucial MX500 1TB 3D NAND SATA M.2 -CT1000MX500SSD4 -- 1TB |
| HDD | SEAGATE 8TB-SATA III-128MB-HDD - ST8000AS0002 -- 8TB |

**Table 2. Experimental results**

| Data set | Automatic blurring process duration [seconds] | Quality control (QC) duration [minutes] | Number of detected objects in automated process [pcs] | Number of detected objects in QC process [pcs] | Percentage of successfully detected objects [%] |
|---|---|---|---|---|---|
| 5000 | 4860 | 105 | 13 478 | 117 | 99.14 |
| 10 000 | 9900 | 170 | 26 152 | 729 | 97.21 |
| 15 000 | 18 240 | 270 | 39 688 | 546 | 98.64 |
| 20 000 | 19 380 | 444 | 52 855 | 845 | 98.43 |

The conducted experiment showed that this methodology is highly precise and is able to perform detection and blurring more efficiently than some other commercially developed software. The software used in this study was tested four times - for four independent panoramic image data sets.

In the first data set, counting 5000 panoramic images, there were 13 478 automatically detected objects in images. During the manual blurring process, i.e. quality control, 117 objects were additionally detected and blurred. Taking this into account, the percentage of successfully blurred objects was 99.14%. This figure was obtained by dividing the number of automatically detected objects with the total number of objects (automatically plus manually detected). In this example, it means 13 478/ (13 478+117) x 100%= 99.14%.

Figure 2 presents the ratio of the number of objects in the images and the total number of images. Here, it can be seen that some images had no objects at all. In this data set, this number was 1143. These include parts of rural area, e.g. forest roads where there were no cars or pedestrians. On the other hand, there are a lot of images counting multiple objects. The maximum number of object in one image was 28.

**Figure 2: Average distribution of objects in images — data set with 5000 images**





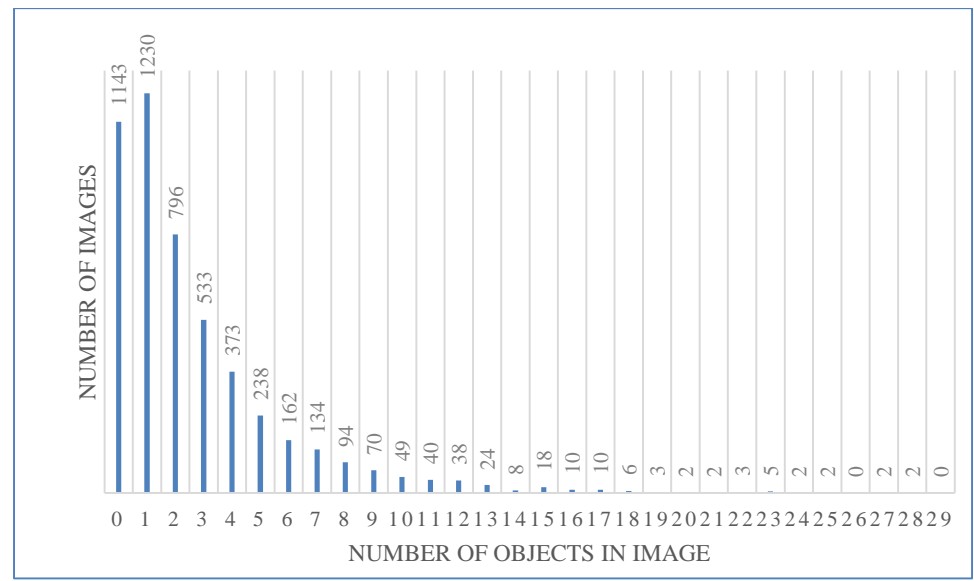


The second data set with 10 000 panoramic images contains 26 152 objects detected by the proposed algorithm and 729 more objects manually found during quality control. The success-percentage equals 97.29%. Figure 3 illustrates a distribution chart of detected objects in this data set. It can be seen that the maximum number of detected objects in one image was 29, while the number of images where there were no bounding boxes was 2371.

**Figure 3: Average distribution of objects in images — data set with 10 000 images**

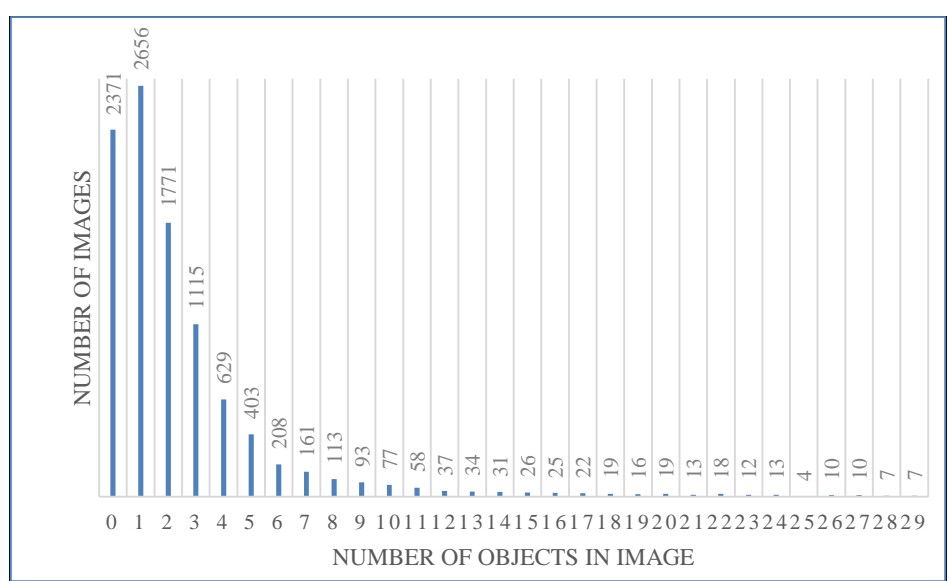

The third data set consists of 15 000 panoramic images. There were 39 688 objects detected and blurred, while 546 objects were found manually. The percentage of successfully blurred images was 98.64%. Figure 4 represents a distribution chart of



detected objects in images of this data set. The highest number of objects detected in one image equals 29, while 3479
images showed no detected objects at all.

**Figure 4: Average distribution of objects in images — data set with 15 000 images**

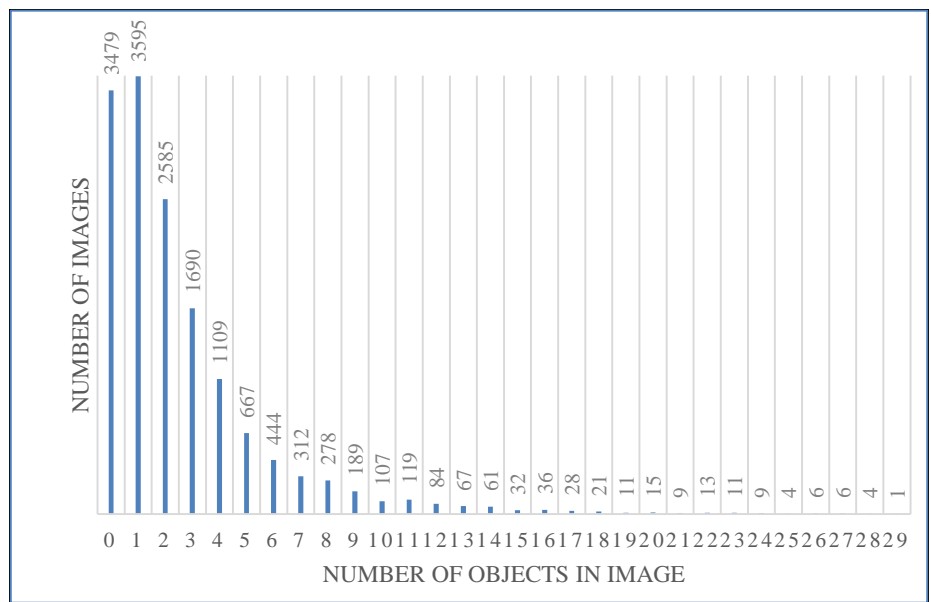

The last tested data set consists of 20 000 panoramic images. The proposed approach produced 52 855 objects which were detected and blurred, while 845 objects were found manually. Figure 5 shows a distribution chart of detected objects in the
fourth image data set, where the maximum number of detected objects in one image was 29, while 4320 images had no objects detected.

**Figure 5: Average distribution of objects in images — data set with 20 000 images**



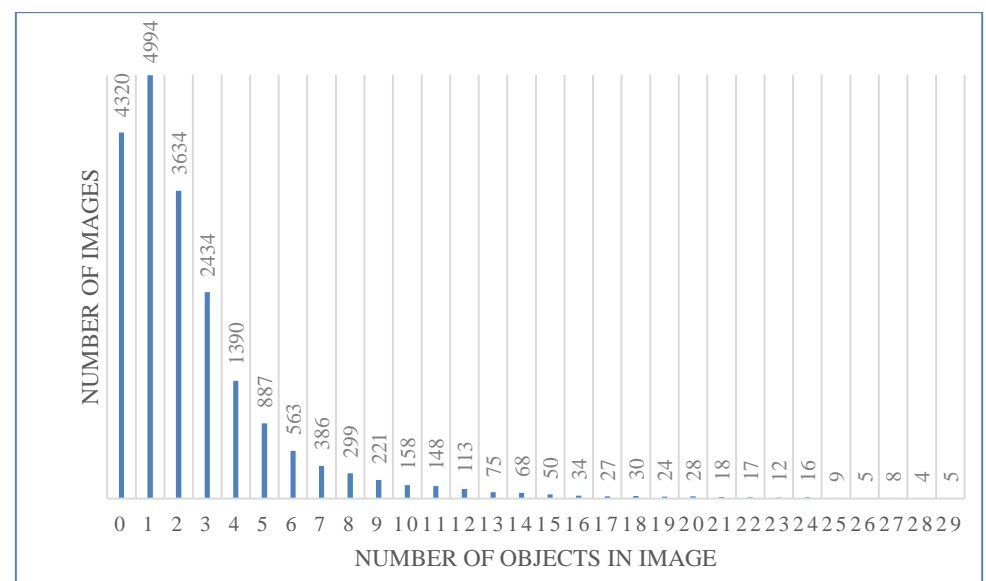

Some examples of positive detected faces and license plates are presented in Figure 6, while some of the false detected or
non-detected objects can be seen in Figure 7.

**Figure 6: Examples of positive blurred panoramic images**

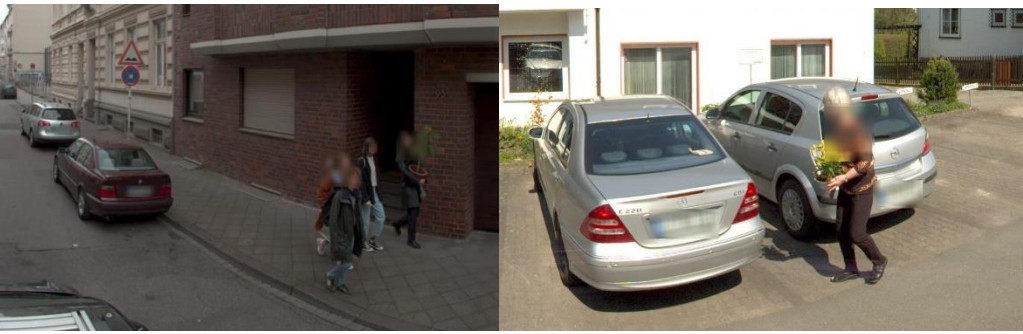

**Figure 7: Examples of negative blurred panoramic images**

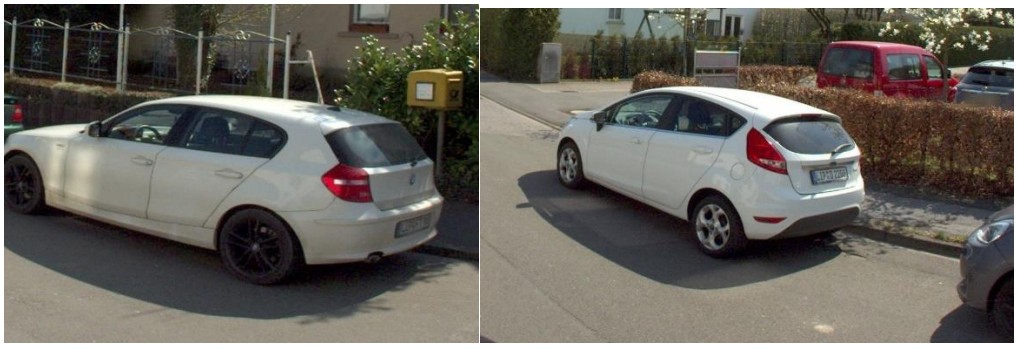


The main limitation of the suggested architecture of the solution for image processing was identified when detecting white cars, because the majority of license plates are white as well. The same color makes it more difficult for the algorithm to execute detection successfully.

**4. Conclusions**

The increasing need for images in mobile mapping projects is highlighted in this paper. Since images are very often used, blurring of faces and license plates is required to comply with data protection laws. According to (Ribaric et al., 2016), privacy is one of the most important social and political issues in contemporary information society, characterized by a growing range of enabling and supporting technologies and services. Amongst these are communications, multimedia, biometrics, big data, rapid development of cloud storage (Wang et al., 2021), data mining, internet, social networks, and audio-video surveillance. Each of these can potentially provide the means for privacy intrusion. Therefore, this article suggests a reliable method for detection and blurring of these particular objects.

The experiment evaluating the applicability of the proposed software was conducted on four different data sets containing urban and rural area panoramic images. The total number of tested panoramic images was 50 000. Those images were obtained from laser scanning of roads with Trimble MX9 device. The success rate results for each data set were as follows: data set with 5000 panoramic images ensured the accuracy of 99.14%, the one with 10 000 panoramic images had 97.21% accuracy, the one comprising 15 000 panoramic images achieved 98.62% and the data set with 20 000 images generated 98.40% of success.

Several major advantages of this approach were identified. Not only that the high percentage of positively detected and blurred elements is evident, but also the presented algorithm was proven remarkably effective in cases when the aforementioned objects of interests had different angles, positions, colors and sizes. Moreover, the usage is very simple since it requires from the user just to put the images that should be blurred into a defined folder and start the blurring machine. Next, its potential is great because it is able to use images of different size, resolution, extension, etc. Finally, its processing speed is extremely convenient as it takes less than one second for detecting and blurring all faces and license plates from one panoramic image.

Future work will involve three focal points. To begin with, the software will be improved in terms of detecting the license plates of the same color as the vehicle they are attached to. Also, since bounding boxes are upright rectangles covering a slightly larger area than the object of interest, working on more precise bounding boxes will be the next stage of further research. They will be marking only the license plate area, taking into account its angle and tilt. In that way, the unnecessary parts of the image will not be blurred, and the quality of the output image in general will be better. Eventually, the plan is to upgrade this methodology with traffic sign detection. Automatic detection of traffic signs with assigned attributes such as



height, sign dimension or type of sign would make a noteworthy contribution to the quality and speed in feature extraction domain.

**Author Contributions:** "Conceptualization, D.V. and M.D.; methodology, I.R.; software, Đ.O.; validation, I.R., M.D. formal analysis, D.V.; investigation, M.D.; resources, D.V.; data curation, Đ.O.; writing—original draft preparation, M.D.; writing—review and editing, I.R.; visualization, D.V.; supervision, Đ.O. All authors have read and agreed to the published version of the manuscript."

**Data Availability Statement.** The data presented in this study are available on FTP server, that can be accessed per
FileZilla. The log-in parameters are as follows: IP: 188.2.72.204, User: Dataset, Pass: Dataset@DD2021, Port: 21, The data could be used only in accordance with GDPR.

**Acknowledgments.** The original data for algorithm testing were provided by DataDEV company, Novi Sad, Republic of Serbia. The paper presents the part of research realized within the project "Multidisciplinary theoretical and experimental research in education and science in the fields of civil engineering, risk management and fire safety and geodesy" conducted
by the Department of Civil Engineering and Geodesy, Faculty of Technical Sciences, University of Novi Sad.

**Conflicts of Interest:** The authors declare no conflict of interest.

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
