# Peer review of "Architecture of Solution for Panoramic Image Blurring in GIS projects Application"

_Geoscientific Instrumentation, Methods and Data Systems, 2021_

## Author Comment (AC2)

[Figure]

Fig 1. Example of 20+ objects detected in one image

---

## Author Response (AR1)

Dear reviewer,

I would like to thank You for having time to read our work. We find Your comments very valuable for improving the paper.

We will try to give answers on all of Your comments.

At first, You mentioned the novelty of our work. The main novelty of this paper is the used architecture. As majority of the approaches for solving the similar problems use just one neural network and search all area of image in order to detect and blur particular elements, we present solution with two or more connected neural networks. Our architecture is designed as a pipeline of object detection. In first step, major regions of interest are detected after which the focus is not on the whole image, but just on the specific parts. That means the proposed solution progressively narrows the search space until it detects the objects to be blurred. This reduced false positives and resulted in high percentage of successfully detected and blurred objects The speed is improved and the method is more effective, comparing to the mentioned existing approaches.

Beside novelty, with this are given answers regarding motivation of our structure. The motivation permeates throughout the paper. This is now emphasised in **abstract** and described into more details in **section 3** (**Figure 1 with more details and paragraph below Figure 1, lines 195-203**), as You proposed.

According to your suggestion, the paper *He, Kaiming et al (2018), Mask R-CNN* is now included in previous research and related work. Also, the diagram showing our architecture of the solution is improved to consist more details about used CNNs (Convolutional Neural Networks), so reader could have better insight into different CNNs and their comparisons (**lines 70-74**).

Then, You mentioned dependency of the used architecture of the process on the result. During creation of this solution, we started with one CNN and result had unacceptable rate of true positives, ie. successfully rate was too low. Beside this, the process took too much time. During the time, we have developed a methodology to achieve a considerable boost in performance of the license plate and face detection algorithms by creating a pipeline that would first localize major regions of interest, such as vehicles and pedestrians, and then pass those regions to the more specialized components of the pipeline that would be tasked with detecting license plates and faces on those smaller regions of interest, effectively simplifying the problem of detection.   Afterwards, it was clear dependency of the used solution on the result.

Verification of our results is done manually (explained in Results and Discussion section), where after automatic process of blurring, operators check each image and mark objects that are not blurred or that are blurred but did not supposed to be. On the basis of these information, statistic is calculated and successfully rate is obtained. Considering the fact that a lot of panorama images are in the city– there are examples of 20+ people and cars at the one image, as presented in Fig. 1 below.

[Figure]

Fig 1. Example of 20+ objects detected in one image

Regarding specific remark: The General Data Protection Regulation (GDPR) is the toughest privacy and security law in the world. GDPR set the firm stance on data privacy and security at a time when more people are entrusting their personal data with cloud services and breaches are a daily occurrence. The mobile mapping data consisting of images needs to be blurred so people can use the data accordingly to the GDPR. We added in **abstract** the short explanation of GDPR.

I hope that these answers meet your criteria and give you a clearer insight into our paper.

Kind regards,

On behalf of all authors

Marina Davidović

Dear reviewer,

I would like to thank You for having time to read our work. We find Your comments very valuable for improving the paper.

Regarding Your suggestion to avoid information repeating, we adjusted the first paragraph in chapter Results and Discussion, so this is corrected now (**lines 210-215**).

You also mentioned characteristics of the computer used for blurring. The idea was not to use super computers and to be focused on the speed (**Results and Discussion section**). We created the architecture of the solution that is effective, available and widely applicable. On the other hand, with improving some components – the process speed will improve, of course. To be more specific, if it is used CPU 20% higher in performances than current, the project management model from our diagram will be improved. If it is used graphic card with higher performances, that will improve object detection modules- vehicle and people and license plate detection modules.

I hope that these answers meet your criteria and give you a clearer insight into our paper.

Kind regards,

On behalf of all authors

Marina Davidović